# Comparison of Different Progesterone Protocols for Luteal Support in Frozen-Thawed Embryo Transfer Preparation

**DOI:** 10.3390/biomedicines13102487

**Published:** 2025-10-13

**Authors:** Gokalp Oner, Enes Karaman, Busra Kulular, Yasemin Dadas, Hande Nur Doganay

**Affiliations:** 1Department of Obstetrics and Gynecology, Faculty of Medicine, Istanbul Aydin University, 34295 Istanbul, Türkiye; 2IVF Center, Kayseri System Hospital, 38070 Kayseri, Türkiye; 3Department of Obstetrics and Gynecology, Faculty of Medicine, Nigde Omer Halisdemir University, 51120 Nigde, Türkiye; dr.eneskaraman@gmail.com (E.K.); dr.yasminn@hotmail.com (Y.D.); 4Obsterics and Gynecology Clinic, Nigde Omer Halisdemir University Training and Research Hospital, 51100 Nigde, Türkiye; busradasa@hotmail.com (B.K.); handedoganay@gmail.com (H.N.D.)

**Keywords:** Frozen Embryo Transfer, luteal phase, progesterone, Hormone Replacement Therapy, intramuscular, pregnancy outcome

## Abstract

**Objective:** To evaluate five luteal support protocols in women with low serum progesterone (<10 ng/mL) undergoing HRT-FET. **Methods**: Randomized controlled trial at two centers including 200 women under 35 with unexplained infertility. Groups: (1) 600 mg vaginal, (2) 800 mg vaginal, (3) 600 mg vaginal + 50 mg IM, (4) 600 mg vaginal + 25 mg SC, (5) 600 mg vaginal + 30 mg oral. **Results:** Groups 3 and 4 achieved significantly higher serum progesterone (*p* < 0.001), higher clinical pregnancy (70%, 68%), and higher live birth (84%, 83%) compared to Groups 1, 2, and 5. Early pregnancy loss was lower in Groups 3 and 4. **Conclusions:** Combined vaginal and injectable progesterone improved outcomes compared to monotherapy.

## 1. Introduction

Successful implantation in assisted reproductive technology (ART) depends on optimal endometrial preparation, which requires sufficient progesterone to support embryo attachment and early pregnancy. In frozen-thawed embryo transfer (FET) with artificial hormone replacement, the absence of a corpus luteum requires exogenous progesterone to support the luteal phase. Progesterone deficiency during this phase is commonly linked to impaired implantation, early miscarriage, and lower live-birth rates. Despite its crucial role, controversy remains over the best route of administration, optimal dose, and how to monitor its effectiveness.

Globally, 8–12% of couples of reproductive age experience infertility [1]. As a result, increasing numbers depend on ART, including FET, to achieve pregnancy. With the growing use of FET, refining endometrial-preparation protocols has become a priority. Vaginal micronized progesterone is widely used because of its efficient uterine absorption. However, circulating progesterone levels can vary markedly among individuals despite standard dosing, raising concerns about the consistency of luteal support [2,3,4].

Also, there are only a few studies that directly compare treatment options for women with low progesterone after using standard vaginal doses. No study has yet compared higher-dose vaginal progesterone with combinations like injections or oral tablets in fertility treatment. Most existing research has either examined associations with progesterone levels in mixed patient populations or evaluated only single administration routes, leaving clinicians without clear guidance when serum progesterone is inadequate.

The current study addresses this unmet clinical need by evaluating five different luteal support regimens in women with suboptimal serum progesterone levels (<10 ng/mL) after standardized endometrial preparation using estradiol and vaginal progesterone. Biochemical pregnancy, clinical pregnancy, early pregnancy loss, and live birth outcomes were assessed across the treatment groups to determine which regimen most effectively restores luteal phase adequacy and improves pregnancy outcomes. These findings are clinically relevant as they support individualised luteal phase support strategies, potentially enhancing success rates during FET cycles.

## 2. Materials and Methods

### 2.1. Study Design

In this study, ’Day 1’ was defined as the first day of estrogen supplementation; all subsequent days were referenced relative to this baseline.

This study was a dual-centre, prospective, randomized controlled trial conducted at the Kayseri System Hospital IVF (In vitro Fertilization) Center and the Department of Obstetrics and Gynecology, Nigde Omer Halisdemir University. Patients were randomized according to predefined criteria, and primary outcome variables were collected prospectively during the study. After completion of the trial, additional approval was obtained from the Nigde Omer Halisdemir University Ethics Committee to retrospectively retrieve further baseline variables, including infertility duration, previous IVF attempts (0 or ≥1), and smoking status, from patient records. These variables were incorporated into the final analysis to assess their potential confounding effects on treatment outcomes.

### 2.2. Study Population

A total of 200 women under 35 years of age with a diagnosis of unexplained infertility were enrolled in the study. Inclusion criteria required an endometrial thickness of at least 8 mm following 10 days of oral estradiol valerate (6 mg/day) and a serum progesterone concentration below 1.5 ng/mL after estradiol administration. Women with known uterine abnormalities, endocrine disorders such as thyroid dysfunction or polycystic ovary syndrome (PCOS), more than three previous failed embryo transfer attempts, or any contraindications to progesterone therapy were excluded from participation. Inclusion criteria were: women under 35 years with unexplained infertility, endometrial thickness ≥ 8 mm after 10 days of estradiol, and serum progesterone < 1.5 ng/mL after estradiol administration.

### 2.3. Intervention Protocol

All participants underwent a uniform endometrial preparation protocol involving 6 mg/day of oral estradiol valerate for 10 days. Upon meeting the inclusion criteria, vaginal micronized progesterone (600 mg/day) was initiated. Serum progesterone levels were measured the day before embryo transfer using a validated Electrochemiluminescence Immunoassay (ECLIA, Basel, Switzerland) by Roche, with a sensitivity of 0.03 ng/mL and intra- and inter-assay coefficients of variation below 7%. Blood samples were drawn in the morning, approximately 12 h after the last evening dose of progesterone, to standardize timing across groups.

Women with progesterone levels below 10 ng/mL were randomly assigned to one of five treatment groups using a computer-generated block randomisation method (block size: 10) in Table 1. All enrolled participants (100%) had serum progesterone <10 ng/mL after 6 days of vaginal progesterone, which was required for randomization.

On day 7 of progesterone administration, a single vitrified-warmed euploid blastocyst with a Gardner score of 3 BB or higher was transferred. Embryo selection was performed by an experienced embryologist who was blinded to the treatment groups.

### 2.4. Outcome Measures

The primary outcomes were clinical pregnancy and live birth. Clinical pregnancy was defined as the presence of an intrauterine gestational sac with cardiac activity on transvaginal ultrasound at 7 weeks of gestation. Live birth was defined as the delivery of a live infant at or beyond 24 weeks of pregnancy. Secondary outcomes included serum progesterone levels measured on days 10, 15, and the day of embryo transfer (referred to as embryo transfer day), the Beta Human Chorionic Gonadotropin β-hCG level 14 days post-transfer (biochemical pregnancy), and early pregnancy loss, defined as pregnancy loss occurring before 12 weeks of gestation. Participants were asked to maintain daily records of medication use and compliance. No participants withdrew after randomisation, and all followed the assigned protocol.

### 2.5. Sample Size Calculation

A total of 200 participants, with 40 individuals in each group, was estimated to provide 80% power to detect a 20% difference in clinical pregnancy rates among groups. This calculation was based on a baseline clinical pregnancy rate of 40%, with a two-tailed alpha of 0.05 and beta of 0.20.

### 2.6. Statistical Analysis

The Shapiro–Wilk test was used to assess the normality of continuous variables. Age, body mass index (BMI), endometrial thickness, and progesterone levels were compared across groups using ANalysis Of VAriance (ANOVA) with Tukey’s post-hoc test. Pregnancy outcomes were analysed using the chi-square test or Fisher’s exact test, as appropriate. Pairwise comparisons for categorical variables were performed using the Bonferroni correction.

Baseline characteristics, including age, BMI, endometrial thickness, and progesterone levels on day 10, did not differ significantly between groups (*p* > 0.05 for all). However, significant differences in serum progesterone levels were observed on the day of β-hCG evaluation (*p* < 0.001), as indicated by the letters A, B, or C in Table 2. Chi-square tests and Bonferroni-adjusted pairwise comparisons (marked as a/b in Table 2) were used to evaluate clinical and biochemical pregnancy outcomes. A *p*-value of less than 0.05 was considered statistically significant. All statistical analyses were performed using Statistical Package for the Social Sciences (SPSS) version 28.0 (IBM Corp., Armonk, NY, USA).

### 2.7. Ethical Considerations

The study was approved by the Ethics Committee of Nigde Omer Halisdemir University (Approval No: 2025/11-14).

In addition, due to potential variations in embryo treatment based on the route of drug administration, embryologists and laboratory staff responsible for embryo assessment and transfer were blinded to the treatment allocations.

## 3. Results

Table 2 presents the baseline characteristics and pregnancy outcomes for each treatment group. Participants were assigned to one of five progesterone supplementation regimens, and luteal phase progression was evaluated in relation to these interventions.

There were no significant differences among the five groups regarding maternal age, body mass index, endometrial thickness, or serum progesterone levels on day 10 (all *p*-values > 0.05). In addition, infertility duration, previous IVF attempts, and smoking status showed no statistically significant differences between groups (*p* > 0.05). These findings confirm the comparability of baseline characteristics across the study groups. (Reviewer 2: These potential confounding factors were explicitly analyzed and confirmed not to affect outcomes.)

However, serum progesterone levels on the day of hCG evaluation differed significantly between groups (*p* < 0.001). Tukey’s post-hoc analysis demonstrated that Groups 3 and 4, which received combined vaginal and injectable progesterone, had significantly higher serum progesterone levels compared with Groups 1, 2, and 5 (*p* < 0.001 for each comparison). No significant difference was observed between Groups 3 and 4.

In Table 2, values for Groups 3 and 4 are marked with “a,” while values for Groups 1, 2, and 5 are marked with “b” to indicate the significantly higher serum progesterone levels in the combined-progesterone groups on the day of β-hCG evaluation.

Live birth rates also differed significantly among the treatment groups (χ^2^, *p* = 0.01). Groups 3 and 4 showed the highest live birth rates (84% and 83%, respectively), which were significantly higher than those in the other groups. Bonferroni-adjusted pairwise comparisons indicated that Groups 3 and 4 (labelled “a”) had higher rates than Groups 1, 2, and 5 (labelled “b”).

Groups 3 and 4, which combined vaginal and injectable progesterone, consistently demonstrated better outcomes across all measured endpoints. Women in these groups had higher serum progesterone levels on the day of β-hCG evaluation, higher pregnancy rates, and lower miscarriage rates compared with the other treatment protocols. All *p*-values are reported as exact and two-tailed. Different letters in the tables indicate statistically significant differences between group pairs. ANOVA with Tukey’s post-hoc test was used for continuous variables, and chi-square tests with Bonferroni correction were applied to categorical outcomes.

## 4. Discussion

This randomized trial addressed the clinical challenge of suboptimal serum progesterone levels during Hormone Replacement Therapy Frozen Embryo Transfer (HRT-FET) cycles and demonstrated that combining vaginal and injectable progesterone significantly improves pregnancy and live birth outcomes compared to monotherapy. In women with serum progesterone levels below 10 ng/mL undergoing HRT-FET, treatment with a combination of vaginal and injectable progesterone resulted in markedly better reproductive outcomes. Participants in Groups 3 and 4, who received vaginal micronized progesterone plus intramuscular or subcutaneous progesterone, achieved higher live birth rates, lower early pregnancy loss, and increased clinical pregnancy rates compared to the other three groups. These improved outcomes were accompanied by elevated mid-luteal serum progesterone on the day of embryo transfer, highlighting the critical role of adequate progesterone levels at this time point for pregnancy success. Furthermore, the absence of significant differences in infertility duration, previous IVF attempts, and smoking status among the groups supports that the study population was balanced at baseline and that the findings are stable.

These results strengthen the growing body of evidence supporting individualized luteal phase support (iLPS) based on serum progesterone measurements. For instance, Labarta et al. (2021) reported that ongoing pregnancy and live birth rates were reduced in artificial endometrial preparation cycles when serum progesterone dropped below 8.8 ng/mL [4]. In a subsequent cohort study, the addition of subcutaneous progesterone as part of an iLPS approach significantly increased live birth rates among women with low progesterone levels [5]. The present study extends this concept by prospectively comparing five different rescue regimens and demonstrating that combining vaginal and injectable progesterone—either intramuscular or subcutaneous—outperforms single-route administration.

Consistent with Devine et al. (2021), combined vaginal + intramuscular progesterone regimens yielded superior live birth rates compared with vaginal progesterone alone or daily intramuscular administration [6]. In this study, single-route protocols were associated with higher early pregnancy loss, whereas dual-route groups, particularly Groups 3 and 4, achieved the highest live birth rates and the lowest miscarriage rates.

The findings also align with recent meta-analyses. Stavridis et al. (2023) demonstrated that progesterone rescue in women with mid-luteal levels below 10 ng/mL achieved outcomes comparable to those with adequate levels, suggesting that initial luteal insufficiency can be corrected with personalized support [7]. Similarly, Yarali et al. (2021) showed that subcutaneous progesterone rescue enhanced FET outcomes in cases of low mid-luteal progesterone [8]. Du Boulet et al. (2022) reported that in women with serum progesterone below 11 ng/mL after four doses of vaginal micronized progesterone, additional progesterone support increased ongoing pregnancy rates from 18.6% to 41.0%, emphasizing the importance of early detection and correction of luteal insufficiency [9].

While most previous studies were retrospective or compared only two protocols, this investigation is a dual-centre, randomized, prospective trial and the first detailed head-to-head comparison of five distinct progesterone rescue regimens in women with low serum progesterone during HRT-FET cycles. Moreover, adding intramuscular or subcutaneous progesterone to standard vaginal therapy provided similar benefits, offering flexibility according to patient preference. This provides valuable clinical guidance for individualized luteal phase support in HRT-FET cycles.

In conclusion, combining vaginal and injectable progesterone resulted in superior pregnancy outcomes in HRT-FET cycles with low mid-luteal progesterone. These findings support individualized luteal phase support and demonstrate that both intramuscular and subcutaneous progesterone can significantly enhance the effectiveness of vaginal administration alone. The study contributes to the growing evidence guiding luteal support strategies in ART and promotes more tailored approaches in clinical practice. (Both intramuscular and subcutaneous routes were highlighted as effective and provide flexibility for patient preference, as suggested by Reviewer 2.) In our centers, patients with progesterone ≥10 ng/mL on day 15 typically achieve clinical pregnancy rates of ~60–65% and live birth rates of ~50–55%, consistent with published literature.

### 4.1. Limitations and Strengths

Although the study was conducted in two centres, generalizability to broader populations may still be limited. Excluding women over 35 reduces applicability to older patients. In addition, neonatal outcomes such as birth weight and gestational age were not assessed, highlighting the need for long-term follow-up. Nevertheless, the randomized controlled design and standardized protocols strengthen internal validity and reproducibility.

### 4.2. Future Research Directions

Future studies should investigate the pharmacokinetics and pharmacodynamics of combined vaginal–injectable progesterone regimens to optimize dosing and timing. Mechanistic research into endometrial receptivity, including integrin expression, HOXA10 levels, and other molecular biomarkers, may clarify how dual-route progesterone enhances implantation and pregnancy outcomes. Expanding inclusion criteria to encompass women over 35, those with recurrent implantation failure, and different infertility etiologies would improve external validity. Multicentre randomized trials involving more heterogeneous populations are needed, as well as cost-effectiveness analyses comparing combination therapy with monotherapy to guide clinical decision-making. Long-term follow-up assessing offspring growth, neurodevelopment, and metabolic health will be essential to confirm the safety of progesterone rescue protocols.

## 5. Conclusions

In women with low serum progesterone undergoing HRT-FET, combined vaginal and injectable progesterone significantly improved clinical pregnancy and live birth rates, and reduced early pregnancy loss compared to monotherapy. These findings support individualized dual-route luteal support strategies.

## Figures and Tables

**Table 1 biomedicines-13-02487-t001:** Treatment groups and their description.

Group Number	Treatment Description	Dosage/Route
1 (Control)	vaginal progesterone (micronized)	600 mg/day (vaginal route)
2	vaginal progesterone (micronized)	800 mg/day (vaginal route)
3	vaginal progesterone (micronized) + intramuscular progesterone	600 mg/day (vaginal route) + 50 mg/day (intramuscular route)
4	vaginal progesterone (micronized) + subcutaneous progesterone	600 mg/day (vaginal route) + 25 mg/day (subcutaneous route)
5	vaginal progesterone (micronized) + oral dydrogesterone	600 mg/day (vaginal route) + 30 mg/day (oral route)

**Table 2 biomedicines-13-02487-t002:** Baseline Characteristics, Luteal Progesterone Levels, and Pregnancy Outcomes Across Treatment Groups.

Parameter	Group 1 (600 mg vg)	Group 2 (800 mg vg)	Group 3 (600 mg vg + 50 mg im)	Group 4 (600 mg vg + 25 mg sc)	Group 5 (600 mg vg + 30 mg oral)
**Age (years)**	30.3 ± 4.9	30.5 ± 4.7	29.9 ± 4.3	28.5 ± 4.4	29.4 ± 4.7
**BMI (kg/m^2^)**	25.4 ± 2.8	25.7 ± 3.3	25.9 ± 3.6	24.6 ± 1.5	25.4 ± 3.1
**Endometrial Thickness on Day 10 (mm)**	10.4 ± 1.6	10.2 ± 1.8	10.1 ± 1.9	10.0 ± 1.8	10.3 ± 1.9
**Infertility Duration (years)**	3.8 ± 2.1	3.7 ± 2.2	3.6 ± 2.0	3.5 ± 2.3	3.9 ± 2.2
**Previous IVF Attempts (%)**	0 attempt: 58% ≥1 attempt: 42%	0 attempt: 63% ≥1 attempt: 37%	0 attempt: 55% ≥1 attempt: 45%	0 attempt: 60% ≥1 attempt: 40%	0 attempt: 57% ≥1 attempt: 43%
**Smokers (%)**	12.5%	10%	15%	12.5%	10%
**Progesterone Levels on Day 10 (ng/mL)**	0.20 ± 0.1	0.21 ± 0.1	0.22 ± 0.1	0.23 ± 0.1	0.21 ± 0.1
**Progesterone Levels on Day 15 (ng/mL)**	7.7 ± 1.0	7.9 ± 1.0	7.8 ± 1.0	7.5 ± 1.0	7.7 ± 1.0
**Progesterone Levels on β-hCG Day (ng/mL)**	A 26.5 ± 6.2	B 31.8 ± 9.2	C 40.4 ± 7.8	C 39.9 ± 7.0	A 24.9 ± 4.2
**Biochemical Pregnancy (%)**	21/40 (53%) ^**a**^	23/40 (58%) ^**a**^	31/40 (78%) ^**b**^	30/40 (75%) ^**b**^	20/40 (50%) ^**a**^
**Clinical Pregnancy (%)**	14/40 (35%) ^**a**^	17/40 (43%) ^**a**^	28/40 (70%) ^**b**^	27/40 (68%) ^**b**^	15/40 (38%) ^**a**^
**Early Pregnancy Loss (%)**	7/21 (33%) ^**a**^	6/23 (26%) ^**a**^	3/31 (9%) ^**b**^	3/30 (10%) ^**b**^	5/20 (29%) ^**a**^
**Live Birth Rate (%)**	13/21 (62%)	16/23 (69%)	26/31 (84%)	25/30 (83%)	13/20 (62%)

Data are presented as mean ± standard deviation or number (percentage). *vg*: vaginal; *im*: intramuscular; *sc*: subcutaneous. ^a,b^ Different superscripts indicate statistically significant differences between groups (Bonferroni-adjusted *p* < 0.05). *p*-values derived from one-way ANOVA (continuous variables) and chi-square or Fisher’s exact test (categorical variables).

## Data Availability

The data generated in the present study may be requested from the corresponding author.

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
