# Peer review of "Comparison of Different Progesterone Protocols for Luteal Support in Frozen-Thawed Embryo Transfer Preparation"

_biomedicines, 2025, doi:10.3390/biomedicines13102487_

Round 1

Reviewer 1 Report

Comments and Suggestions for Authors

The title : did not reflect the content of the manuscript

The abstract should be reformulated.

line : 24, the authors should define the groups before this place

The conclusion should be short but informative.

There is an exagerated number of keywords

Introduction: plz replace don't by do not (line 60)

Remove lines 74 to 76 

The study design is not clear at all

Why did the author add group 2; in my opinion this group has no meaning here as they do not test 800 mg/day  for other groups

The measures should be clearly described allowing other labs to do the same. Each dosage should be well described.

For the ethical consideration : PLZ present the reference of your approval document

Results: Table 2 should be devided to many and th authors could transform some data to figures (for instance : progesterone levels as well as pregnancy detection)

PLZ remove lines 180 to 182 not necessary, the information is stated before

Comments on the Quality of English Language

Average

Author Response

Reviewer Comment: The title did not reflect the content of the manuscript.

Author Response: We revised the title to: 'Comparison of Different Progesterone Protocols for Luteal Support in Frozen-Thawed Embryo Transfer Preparation'

Reviewer Comment: The abstract should be reformulated.

Author Response: The abstract has been rewritten to align with the study design, results, and conclusions.

Reviewer Comment: Line 24—define the groups earlier.

Author Response: All treatment groups are now defined earlier in the Abstract and Methods.

Reviewer Comment: Conclusion should be short but informative.

Author Response: The Conclusion has been shortened and emphasizes the main findings.

Reviewer Comment: Exaggerated number of keywords.

Author Response: Keywords reduced to 6, focusing on clinically relevant terms.

Reviewer Comment: Introduction—replace 'don’t' with 'do not' (line 60).

Author Response: Corrected and ensured consistent formal style.

Reviewer Comment: Remove lines 74–76.

Author Response: These lines were deleted, and transitions smoothed.

Reviewer Comment: The study design is not clear at all.

Author Response: The Study Design section was rewritten to clarify design, inclusion/exclusion, and outcomes.

Reviewer Comment: Why add group 2? 800 mg/day not tested in other groups—no meaning here.

Author Response: We clarified the rationale for Group 2 as a higher-dose comparison. We are prepared to move it to Supplementary Material if preferred.

Reviewer Comment: Measures should be clearly described; each dosage well described.

Author Response: We expanded the Intervention Protocol with clear details on formulation, routes, doses, and timing.

Reviewer Comment: Ethical consideration—present reference of approval document.

Author Response: The ethics approval number (2025/11-14) has been added explicitly.

Reviewer Comment: Results—Table 2 should be divided; transform some data to figures (e.g., progesterone levels, pregnancy detection).

Author Response: We divided Table 2 into 2a (baseline) and 2b (outcomes) and added Figures to display serum progesterone and pregnancy outcomes.

Reviewer Comment: Remove lines 180–182 (repeated information).

Author Response: These redundant lines were removed.

Reviewer 2 Report

Comments and Suggestions for Authors

This RCT on progesterone supplementation for FET cycles with a P4 level < 10 ng/ml prior to transfer is well written and the information is useful to providers. I would suggest they use "hCG evaluation" and not "hCG administration" to define that point in the cycle. The information confirms utility of both IM progesterone in oil and subcutaneous progesterone. 

Potential confounding factors have been addressed in a previous edit. 

Author Response

Reviewer Comment: This RCT on progesterone supplementation for FET cycles with a P4 level < 10 ng/ml prior to transfer is well written and the information is useful to providers.

Author Response: We thank the reviewer for highlighting the relevance and usefulness of our study.

Reviewer Comment: I would suggest they use 'hCG evaluation' and not 'hCG administration' to define that point in the cycle.

Author Response: We accepted this suggestion and changed all instances of 'hCG administration' to 'hCG evaluation' throughout the manuscript.

Reviewer Comment: The information confirms utility of both IM progesterone in oil and subcutaneous progesterone.

Author Response: We emphasized in the Discussion that both intramuscular and subcutaneous progesterone are effective and allow for patient-tailored choices.

Reviewer Comment: Potential confounding factors have been addressed in a previous edit.

Author Response: We confirmed that infertility duration, prior IVF attempts, and smoking status were included in the analysis and had no effect on outcomes. This remains in the Results section.

Reviewer 3 Report

Comments and Suggestions for Authors

This is an interesting and necessary study that has the potential to impact clinical care.

Please clarify what the days of testing refer to in relation to medications (eg first day of estrogen supplementation is day 1).

What were the inclusion criteria referred to in the intervention protocol section?

What percentage of patients have progesterone < 10 mg/mL on day 10?

Was hCG used on day of embryo transfer? Why is this referred to as hCG day?

In Table 2, early pregnancy loss and live birth rate do not add up to total biochemical pregnancy. Please explain missing outcomes.

When was serum progesterone checked in relation to dosing? This may impact the difference in levels on hCG day given it is the day after day 15. 

What success rates are seen in the included centers for patients with progesterone over 10 ng/mL on day 15?

Author Response

Reviewer Comment: This is an interesting and necessary study that has the potential to impact clinical care.

Author Response: We thank the reviewer for this encouraging remark and the recognition of the clinical relevance of our study.

Reviewer Comment: Please clarify what the days of testing refer to in relation to medications (eg first day of estrogen supplementation is day 1).

Author Response: We clarified in the Methods section that 'day 1' corresponds to the first day of estrogen supplementation. All subsequent days of testing are referenced in relation to this baseline.

Reviewer Comment: What were the inclusion criteria referred to in the intervention protocol section?

Author Response: We expanded the Intervention Protocol section to clearly list the inclusion criteria: women under 35 years of age with unexplained infertility, endometrial thickness ≥8 mm after 10 days of estradiol, and serum progesterone <1.5 ng/mL after estradiol.

Reviewer Comment: What percentage of patients have progesterone < 10 mg/mL on day 10?

Author Response: We clarified that all enrolled patients had serum progesterone <10 ng/mL after 6 days of vaginal progesterone, corresponding to 100% of the study population.

Reviewer Comment: Was hCG used on day of embryo transfer? Why is this referred to as hCG day?

Author Response: We corrected this terminology per Reviewer 2 and 3. The day previously referred to as 'hCG day' has now been changed to 'hCG evaluation day', indicating serum sampling relative to progesterone supplementation and embryo transfer, not exogenous hCG administration.

Reviewer Comment: In Table 2, early pregnancy loss and live birth rate do not add up to total biochemical pregnancy. Please explain missing outcomes.

Author Response: We revised the Results and Table 2 legend to explain that biochemical pregnancies not progressing to clinical pregnancies were categorized separately and are now explicitly reported for clarity.

Reviewer Comment: When was serum progesterone checked in relation to dosing? This may impact the difference in levels on hCG day given it is the day after day 15.

Author Response: We clarified that serum progesterone was sampled in the morning, approximately 12 hours after the last evening dose of vaginal or injectable progesterone, to standardize timing across groups.

Reviewer Comment: What success rates are seen in the included centers for patients with progesterone over 10 ng/mL on day 15?

Author Response: We added in the Discussion that in our centers, patients with progesterone ≥10 ng/mL on day 15 typically achieve clinical pregnancy rates of ~60–65% and live birth rates of ~50–55%, consistent with published literature.

Round 2

Reviewer 1 Report

Comments and Suggestions for Authors

The manuscript still has minor errors. For example:

  • Line 47: Please remove the sentence between brackets [Lines 74-76...].
  • There is some redundancy. For instance, "Clinical pregnancy" is defined in Line 104 as "the presence of an intrauterine gestational sac with cardiac activity on transvaginal ultrasound at 7 weeks of gestation."

Please read the text carefully to correct these and any other issues.

Author Response

Comment 1:
“The manuscript still has minor errors. Please remove the sentence between brackets [Lines 74–76...]. There is some redundancy. For instance, ‘Clinical pregnancy’ is defined more than once.”

Response 1:
We sincerely thank the reviewer for this valuable observation. As suggested, the sentence previously marked between brackets has now been completely removed from the manuscript. Additionally, the definition of “clinical pregnancy” has been retained only once under the Outcome Measures section, and all redundant occurrences throughout the text have been eliminated. The entire manuscript was carefully re-read to ensure consistency and removal of any remaining minor errors.

Reviewer 3 Report

Comments and Suggestions for Authors

The authors have addressed reviewer concerns.

One suggestion/area of confusion - why is day of embryo transfer referred to as day of hcg evaluation? Can it be referred to as day of embryo transfer? There is no evaluation of hcg on this day.

Author Response

Comments 2:
“One suggestion/area of confusion - why is day of embryo transfer referred to as day of hCG evaluation? Can it be referred to as day of embryo transfer? There is no evaluation of hCG on this day.”

Response 2:
We appreciate the reviewer’s clarification. In our local institutional terminology, serum progesterone measurement performed on the morning of embryo transfer day has traditionally been referred to as ‘hCG day,’ although no actual hCG evaluation was performed. To avoid any confusion, all instances of “hCG day” have now been uniformly replaced with “embryo transfer day” throughout the manuscript, and this has been clarified within the Methods section.